# Dynamic Environmental Physical Cues Activate Mechanosensitive Responses in the Repair Schwann Cell Phenotype

**DOI:** 10.3390/cells10020425

**Published:** 2021-02-17

**Authors:** Michele Fornaro, Dominic Marcus, Jacob Rattin, Joanna Goral

**Affiliations:** 1Department of Anatomy, College of Graduate Studies (CGS), Midwestern University, Downers Grove, IL 60515, USA; jgoral@midwestern.edu; 2Department of Anatomy, Chicago College of Osteopathic Medicine (CCOM), Midwestern University, Downers Grove, IL 60515, USA; dmarcus46@midwestern.edu (D.M.); jrattin76@midwestern.edu (J.R.)

**Keywords:** Schwann cells, myelination, demyelination, nerve injury, environmental cues, mechanosensitivity, extracellular matrix, repair phenotype, cell plasticity

## Abstract

Schwann cells plastically change in response to nerve injury to become a newly reconfigured repair phenotype. This cell is equipped to sense and interact with the evolving and unusual physical conditions characterizing the injured nerve environment and activate intracellular adaptive reprogramming as a consequence of external stimuli. Summarizing the literature contributions on this matter, this review is aimed at highlighting the importance of the environmental cues of the regenerating nerve as key factors to induce morphological and functional changes in the Schwann cell population. We identified four different microenvironments characterized by physical cues the Schwann cells sense via interposition of the extracellular matrix. We discussed how the physical cues of the microenvironment initiate changes in Schwann cell behavior, from wrapping the axon to becoming a multifunctional denervated repair cell and back to reestablishing contact with regenerated axons.

## 1. Introduction

The outcomes of peripheral nerve repair are highly variable and highly dependent on many factors, including the extent and type of injury. A peripheral nerve is made of axons surrounded by Schwann cells (SCs) and connective tissue component where blood vessels can be found [1]. Two types of SCs can be found in a peripheral nerve. The first type, myelinating SCs, generate the myelin sheath made of multiple wrapping layers of plasmalemma around axons. The SC plasma membrane has a high lipid content, with cholesterol being especially important for myelin sheath assembly [2,3]. The second type, non-myelinating Remak SCs, surround nerve axons without forming the myelin sheath. A single myelinating SC myelinates a single axon, while a single Remak SC is associated with several, typically smaller, axons (<1 µm in diameter), forming an assembly called the Remak bundle [4]. Both myelinating SCs and Remak SCs coexist in a peripheral nerve performing their distinctive roles to maintain the proper function of a nerve. Importantly, both types of SCs play key roles in a peripheral nerve repair. Disruption of the normal myelination process, which could happen as a result of nerve injury, can lead to a higher proportion of axons associated within Remak bundles, potentially causing abnormal hypersensitivity to stimuli due to a higher fiber density in Remak bundles [5].

In the event of a peripheral nerve injury, both types of SCs change their phenotype and adopt a new role as repair cells. A dramatic transformation will lead to dedifferentiation, increase in proliferation and migration into the lesion site, where repair SCs will create a permissive environment for the injured axons to regenerate.

Both chemical and physical signals contribute to the transition from mature to the repair SCs. Historically, the majority of studies investigated intracellular signaling pathways activated or downregulated in SCs as a result of nerve damage [6]. More recently, the significance of physical cues coming from the microenvironment at the site of injury in which SCs acquire the repair phenotype have started to be examined. Specifically, it was reported that SCs are able to modify their behaviors by sensing mechanical properties of their microenvironment, such as stiffness and elasticity. This property is designated as mechanosensitivity, is a key factor not only in the proper development of a peripheral nerve but, also, in its maintenance and repair [7,8,9,10]. That is, SCs respond to the physical constraints at a molecular level by mechanotransduction, the induced activation of genetic programs and cytoskeletal remodeling in response to physical stimuli [11,12,13]. This new perspective proposed and adopted in recent studies recognizes the importance of the microenvironment at the site of nerve injury. In this new approach, biophysical cues initiate and lead to biochemical changes in the microenvironment that ultimately result in the adaptive reprogramming defining the phenotype of the repair SCs [14,15,16].

In this review, we aim to describe different stages—the result of changes in the environmental conditions—that characterize the process of transformation of SCs from the mature differentiated to the repair phenotype. Our goal is the reinterpretation of the process of SC reprogramming during nerve repair as a consequence of signals initiated by alterations in the microenvironment. In turn, these environmental cues affect multiple intracellular signaling pathways.

We identified four different microenvironments that SCs will encounter during their transformation into the repair phenotype. We will discuss how the SC phenotype changes in response to their ability to sense physical stimuli that characterize these four different microenvironments. The first phase is initiated by the injured axon degeneration. SCs that are in intimate contact with these axons sense the mechanical changes at the SC–axon interface and start the transformation from mature enwrapping to denervated SCs. The second phase, the fluid phase, is characterized by the local accumulation of plasma exudate and recruitment of macrophages and activated SCs at the lesion site. During this phase, the lesion is cleared from the cellular and myelin fragments [17]. The next phase is the matrix phase. This phase is characterized by the deposition of the extracellular matrix (ECM), which is filled with fibrils that influence a directional migration and alignment of SCs to control axonal growth. During the final phase, the presence of regenerated axons stimulates the differentiation of SCs into the mature phenotype. During this phase, environmental cues are conducive to remyelination.

## 2. Phase One: Detachment from Degenerating Axons—Denervation

Unlike oligodendrocytes, which are the functional counterparts of SCs in the central nervous system, SCs are immersed in connective tissue with the basal lamina at the interface of SCs and connective tissue ECM. During development, myelination is controlled by interactions between axons, SCs and the basal lamina in the process known as radial sorting [18]. It is possible that the same players are involved in the demyelination of an injured nerve. A physical injury to the nerve affects the SC–axon interface and forces the detachment of SCs. Moreover, the injury causes changes in the surrounding ECM. Both myelinating and non-myelinating SCs react to the injury and undergo an identity change to mount a response to the injury and, then, to support nerve regeneration. Within 24 h of the lesion, the axoplasmic cytoskeleton disintegrates and myelin breaks down into small segments (Figure 1) [19].

Denervated SCs respond to the deteriorating conditions of resected axons regressing to a dedifferentiated state similar to the neonatal SC progenitor cells. After this initial regression to the progenitor-like cell, SCs activate different molecular mechanisms to eventually transform into the regeneration-supporting phenotype. Through this process, SCs will acquire the new identity of repair SCs [10,20,21,22,23,24]. The regression to a progenitor-like cell phase is characterized by a reversal of all developmental steps that assist SCs during the functional acquisition of tools for proper interactions with axons. This ability of SCs to remodel themselves morphologically and functionally in response to traumatic stimuli was described by Jessen et al. as “adaptive cellular reprogramming” [1,13,16]. The reprogramming is driven by a combination of dedifferentiating and differentiating programs. Dedifferentiation involves the downregulation of pro-myelin factor Krox20 and its downstream signaling pathways, which leads to the decreased synthesis of structural proteins such as P0, the myelin basic protein (MBP) and myelin-related membrane proteins such as myelin associated glycoprotein (MAG) and periaxin [25,26,27]. The cholesterol synthesis is also downregulated. Not surprisingly, dedifferentiation of the mature SC phenotype is accompanied by the upregulation in the transcription of a number of proteins—including Sox-2, Pax-3, Notch, L1, NCAM and GFAP—which characterize immature SCs [13]. The same primary regulators for adaptive cellular reprogramming are also activated in non-myelinating Remak SCs. However, in contrast to neonate SCs, denervated SCs maintain a high profile of N-cadherin and integrin α1β1 [28,29,30] and express de novo molecular markers like Sonic Hedgehog (Shh) and Olig1, which are normally absent during development [31,32,33].

The major regulator of SC adaptive cellular reprogramming into the repair phenotype is the transcription factor c-Jun [34,35,36]. In mice with inactivated c-Jun in SCs, the repair processes were fundamentally disrupted, resulting in a lack of activation of the repair program and myelin clearance. The downstream signaling pathways activated by c-Jun or identified as affected in the c-Jun mutant mice involve the activation of other transcription factors and the production of cell-adhesion molecules and growth factors important in the repair program [20,31,34,37,38,39,40,41]. The regulation of the c-Jun pathway is highly dependent on the physical conditions that regulate the microenvironment surrounding SCs.

Recently, an in vitro study showed a direct correlation between the activation of c-Jun expression in SCs and increase in the content of laminin in the growing substrate. This finding suggests that a high concentration of laminin may characterize the in vivo ECM environment of the initial phase after nerve injury, in which dedifferentiated SCs separate from the axons [10]. The maintenance of myelination in the presence of normal contact with axons is regulated by signaling pathways, including the hippo pathway, known to balance physical and chemical cues that regulate SC behavior. The downstream effectors of this pathway, YAP/TAZ, inhibit Notch signaling and downregulate c-Jun expression [42,43,44]. This process is reversed when the physical contact between SC and the axon is interrupted. In response to mechanical signals from the ECM, YAP/TAZ shuttle into the nucleus and activate the transcription of basal lamina receptor genes, including laminin receptors [11,12].

Many studies affirmed the role of neuregulin 1 (NRG-1), especially its type III isoform, and its interaction with tyrosine kinase receptor heterodimer ErbB2/3 in the myelination and non-myelination processes during development as it relates to the axonal size, also known as radial sorting [5,45]. NRG-1 levels in the mature peripheral nerves are very low; however, its expression is induced in SCs after a nerve injury (see also Phase Four: Contact with Regrowth Axons). High levels or an overexpression of NRG-1 are linked to demyelination and may play a role in chronic demyelinating neuropathies [46,47,48]. In a purified Schwann cell culture, Cheng et al. [49] demonstrated a suppression of P0 in correlation with a high level of NRG-1. Moreover, when SCs are cocultured with neurons, high levels of NRG-1 inhibit myelination and promote demyelination [50,51]. Intracellular ErbB2/3 signaling activates the MAPK/ERK signaling cascade, which has been reported to suppress myelination [52,53].

Additional evidence suggests a correlation between the activation of ERK1/2 and demyelination via a direct link between the increased expression of ERK1/2 and increased levels of the chemokine MCP-1/CCL2, which recruits macrophages to the site of lesions [54]. Both reprogrammed SCs and newly arriving macrophages begin to remove cellular debris at the lesion site by phagocytosis. They also produce proinflammatory cytokines, such as TNFα, LIF and MCP-1/CCL2, and regulators of myelination, such as Gpr126 [55,56,57].

## 3. Phase Two: The Fluid Phase

This phase is critical for SC proliferation and sets the stage for cell migration at the lesion site (Figure 2). It is a transient and relatively short phase (24–48 h) between the deconstruction of the old nerve structure and reconstruction of the new. Belkas designated this phase as the “fluid” phase [17]. This phase is characterized by the accumulation of plasma exudate and increased levels of chemokines and growth factors. As a result, multiple cell types will migrate into the injured nerve to set the stage for regeneration [58,59,60].

The remarkable event that starts this phase is myelinophagy, which corresponds to the Wallerian degeneration at the site of axon lesion [19,61,62,63,64,65]. Within hours, the immune response driven by upregulation of the proinflammatory cytokines is activated. As SCs disassociate from degenerating axons, they adopt the macrophage-like phagocytic phenotype and start producing macrophage chemoattractants. With this new role, repair SCs start the phagocytosis of the content of the nerve stump. They also produce prodegenerative molecules like SARM1 and PHR1 [65]. Macrophages migrating to the site of nerve injury join SCs in the myelinophagy process.

By actively producing neurotrophic factors and proteins into the ECM, the reprogrammed SCs change the ECM composition and create a new environment that promotes regeneration of the severed axons [66]. In vitro data show a peak in the concentration of neurotrophic factors within hours post-lesion [67]. The upregulation of these factors occurs in an orderly fashion starting with the neurotrophins family, followed by the GDNF family neurotrophic factors and, lastly, the neuropoietic cytokines [68].

The nerve growth factors identified at the lesion site and their pro-survival effects have been extensively studied in the field of neural scaffolds used in nerve repair [69,70,71,72,73,74,75,76,77]. Various designs of nerve conduits and integrated delivery systems of growth factor have been implemented. The studies involved the evaluation of releasing properties of the conduit walls with the use of biological tissues or synthetic biomaterials. Additionally, different type of matrices or cells—such as SCs, stem cells and mesenchymal cells—are assessed to fill the conduit. Another broad area of study evaluates the optimal ways of growth factor delivery into the lumen of the nerve conduit. Such research extends from the genetic reprogramming of endogenous cell types to the implementation of microspheres or physical or chemical cues to topically deliver growth factors [78].

The high content in neurotrophic factors in the fluid phase microenvironment promotes the proliferation of SCs and the upregulation of pro-survival pathways [79,80,81]. For instance, the upregulation of cell surface receptors like P2X7 and an ATP-driven ligand-gated Ca^2+^ channel activates the MAPK/ERK pathway [82]. Importantly, activation of the Raf-MAPK/ERK and PI3K-mTOR signaling pathways controls demyelination, inflammatory responses and maintenance of the SC repair phenotype [56]. Ultimately, the ERK pathway regulates the c-Jun pathway, which plays a major role in SC reprograming [52].

Although SCs are considered essential for nerve regeneration, other cell types that are present at the site of the lesion are also active in the process of nerve repair. The results of several in vivo studies indicated that the area between the proximal and distal stump in a rat model of a peripheral nerve injury contained mostly macrophages (50%), followed by neutrophils (24%), fibroblasts (13%) and endothelial cells (5%). The number of endothelial cells increased gradually, possibly as a result of the revascularization of the nerve at the site of the lesion [58,59,60]. This was supported by the evidence that macrophages at the site of the lesion produce the angiogenic factor VEGF-A in response to a hypoxic environment [60]. Among the cell types present at the site of injury, macrophages are the only cells with such characteristics. The angiogenic role of macrophages as a source of VEGF in the hypoxic environment of developing tumors is well-documented [83,84]. In a positive-feedback mechanism, high levels of VEGF-A further recruit macrophages to the site of injury [85]. Therefore, hypoxia is another important biophysical stimulus characteristic of the evolving environmental conditions at the site of the lesion at this stage. High levels of VEGF-A activate the proliferation and migration of endothelial cells into the site of the lesion. Thus, angiogenesis is one of the important steps associated with the evolving conditions of the microenvironment characterizing the fluid phase [60]. The invasion into the lesion site by SCs will follow. Since angiogenesis at the site of the nerve lesion is initiated by macrophages and executed by endothelial cells via the VEGF-A pathway, it is evident that this final stage of the fluid phase is mostly driven by cells other than SCs. Therefore, the other cell types associated with nerve injury prepare the environment for SC migration to the lesion site to continue the nerve regeneration process. Once at the lesion site, SCs will then actively secrete and release fibrils, further changing the composition of the ECM.

## 4. Phase Three: The Matrix Phase

Following the high levels of SC proliferation characteristics of the fluid phase, the next essential step is their migration. Repair SCs migrate in two opposite directions from both the proximal and distal stumps (Figure 3). The environmental conditions of the migratory pathway lack uniformity and constantly change during regeneration. This presents a particular challenge, since the success of nerve repair is highly dependent on the ability of SCs to effectively migrate across the injury site [9].

Along the migratory pathway, SCs form aligned elongated tubular structures called bands of Bungner that provide guidance to regrowing axons and support their linear regeneration [16,86]. A failure of SCs to migrate across the lesion area or the lack of alignment and guidance to growing axons may affect the process of nerve regeneration [9]. Therefore, the efficiency of nerve regeneration depends on both the ability of SCs to migrate and to maintain the directionality of the migratory path. These functional properties of SCs are highly dependent on the type of ECM substrate and its physical variability.

The important aspects that contribute to proper directional migration are the morphological changes that confer a promigratory phenotype to SCs [36,43,87,88,89] and the microenvironmental cues at the lesion site that orient migrating SCs [7,8,9,90,91].

In this section, we will describe the environmental cues that activate changes in the morphology of the promigratory SC and identify the physical cues that confer directionality to SCs during their migratory path across the site of the lesion.

Promigratory repair SC cells have a different morphology from the SCs that are in contact with axons. The repair SCs are elongated and have branches. While migrating, they start aligning in a bridge-like fashion across the area between stumps. The elongation allows for a maximal overlap between the repair SCs to ensure a continuous, uninterrupted track for the regenerating axon [36].

In this phase, an active secretion and deposition of oriented fibrin in the ECM provides an increasingly stiffer substrate that allows for directional guidance for migratory SCs. At this time, the environment gradually changes from a fluid-filled to fibril-filled matrix. The formation of this fibrin matrix, although critical for the nerve regeneration [92], does not seem to be essential for the sprouting and elongation of axons. However, the formation of the fibrin matrix is important in creating the lining substrate for migrating SCs to populate the new environment in anticipation of the axons. ECM fibrils are secreted by fibronectin-producing repair SCs. [93,94]. Numerous fibroblasts present at the site of the lesion also synthesize fibronectin. It was reported that SCs treated with glial growth factor (GGF) in primary cultures increase the production of fibronectin [87]. In addition, a GGF-dependent increase of fibronectin production was associated with an increase in SC proliferation and favored SC invasion into the area of injury [95].

The active secretion and deposition of fibrils in the matrix along the degenerating/regenerating nerve creates an increasing gradient of stiffness along the SC migratory path. This gradient plays an important role in eliciting the promigratory and pro-regenerative phenotype of SCs [8,12,91,96,97]. The change in the substrate stiffness is one of the physical cues that contributes to the lack of uniformity of the migratory path. SCs sense the gradient of stiffness and orient their movement accordingly. This property of the microenvironment that affects the direction of migration and the morphology of repair SC is known as durotaxis [98]. The same growing stiffness of the substrate favors directional axonal growth [7]. Evans et al. [9] recently reported that, unlike uniform substrates at different degrees of stiffness, a substrate with a gradient of stiffness can induce morphological changes in migrating SCs. Moreover, SCs grown on a steep gradient are significantly bigger along both cellular axes and feature an increased number of complex lamellipodia. These cells also form numerous focal adhesions at the plasma membrane that are important for SC migratory behavior [99,100]. Furthermore, time-lapse microscopy revealed that the shape of the cells changed in correlation with the type of substrate. While cells grown onto a uniform substrate are bipolar, their morphology becomes more complex on a gradient substrate with a dominance of a multipolar morphology displaying extensive branching. In addition, the velocity and directedness of SC migration increases on gradient compared to uniform substrates [9].

The mechanotransduction response to the physical changes in the stiffness of the substrate involves secretion of the nerve growth factor (NGF), brain-derived neurotrophic factor (BDNF) and ciliary neurotrophic factor (CNTF) into the ECM. This further changes the ECM composition, creating an environment suitable for axonal growth and the reestablishment of SC–axon interactions [91,97]. The stiffness of the substrate has also been correlated with a fluctuation in the production of adhesion molecules such as β-catenin and N-cadherin, which are important for migration and anchoring to the substrate [97].

Stiffness is one of the physical properties of the substrate that can affect the directional orientation of SCs. Migratory activities of various types of cells in vivo involve migration over a cell-made substrate and depend on the types of interacting cells. For example, endothelial cells migrate along arteries during development [101]. Additionally, in adult neurogenesis, neuroblasts can migrate along blood vessels [102]. Recent data showed that SCs may also benefit from the presence of other cell types. Moreover, live tissue at the site of injury may facilitate their migration and define a directional path. For example, it was reported that fibroblasts present at the lesion site seem to affect SC directional migration. Fibroblasts share with SCs the ability to secrete fibronectin and, therefore, contribute to the production of ECM fibrils. Fibroblasts have also been reported to induce morphological changes in migratory SCs and their adhesive behavior via EphrinB/EphB2 signaling [60,103]. In the facial nerve injury model, the Shh signaling pathway has been shown to affect the action of dormant fibroblasts present at the injury site. Specifically, Gli1+ fibroblasts were involved in rebuilding the nerve sheath in facial nerve regeneration via the production of fibronectin. The presence of Gli1+ fibroblasts in the scaffolds bridging the proximal to distal stump was considered essential for proper nerve regeneration [104].

Another tissue-like substrate considered essential for SC migration is defined by the presence of newly sprouted blood vessels at the site of the lesion. An increased concentration of VEGF-A produced by macrophages at the site of injury promotes the new formation of blood vessels. It was reported that these vessels grow in a polarized manner across the injured nerve [60]. Notably, reorienting the sprouting blood vessels or inhibiting angiogenesis compromised the directional migration of SCs [60]. The observation that SCs may sense the orientation pattern of blood vessels while migrating and aligning to form the tunneling bands of Bungner provides insights into the importance of environmental physical cues that contribute to a successful regeneration. It also validates the importance of topographic patterned substrates for nerve repair. For instance, the application of tissue engineering techniques to formulate suitable surfaces to use as the nerve scaffolds demonstrates the importance of topography for SC orientation. Specifically, SCs oriented themselves parallel to laser-fabricated and elliptically shaped microconical surfaces or exhibited random orientation when grown onto an arbitrarily shaped topography [105]. In addition, the same study demonstrated that the topography affected the orientation of the growth of sympathetic neurons axons. Similar results were reported from studies showing how micropatterned substrates of different geometrical characteristics affect SC adhesion, proliferation and orientation [10,106,107,108].

## 5. Phase Four: Contact with Regrowth Axons

The new physical cue characterizing this final phase is the presence of regenerated axons (Figure 4). Pulled by the SCs migrating from the proximal stump, axons repopulate at the site of a lesion, growing onto the tubular-like bands of Bungners and forming contacts with the SCs guiding their path. The physical contact with the axon will induce new morphological changes in the repair SCs to plastically adapt and reestablish the lost cell–axon relationship. The repair SCs have different sizes and morphologies than mature myelin SCs and Remak SCs. The contact with the axon reversibly decreases the size of the repair SCs to about one-seventh [36].

The SC is a polarized cell characterized by two membrane surfaces: the adaxonal surface opposing the axolemma and the abaxonal surface in contact with the connective tissue ECM environment via its basal lamina [8]. A combination of signals from the abaxonal surface, which is mechanosensitive to the fibrillary composition of the ECM, generates counterpart signals at the adaxonal surface of the SC. These signals implement adaptive programs that ultimately result in reorganization of the cytoskeleton at the interface with the axon [8]. The new physical stimuli lead to genetic reprogramming and new signaling patterns. The same pro-myelin factors that were downregulated in phase one because of the loss of the axons are now upregulated by the contact with the axolemma. Numerous receptor-mediated intracellular signaling pathways that regulate cell–cell interactions have been identified [6]. Importantly, it was recently demonstrated that GPR126, the mechanosensitive receptor that plays a role in demyelination, is also active in the remyelinating process [8]. Activated by mechanical stimuli perceived from the microenvironment, GPR126 promotes the pro-myelin transcription factor Krox20. Moreover, the concentration of Krox20 increases proportionally with the increase of stiffness of the ECM [8].

It was also reported that another mechanosensitive mechanism involving YAP/TAZ was transiently activated by changes in the ECM mechanical properties that regulate the SC–axon interactions. This observation further supports the concept that environmental physical cues may condition SC behaviors [10,11,12].

A morphological analysis that compared a degree of myelination before or after a lesion revealed that the normal ratio between the diameter of the axon and myelin sheath thickness was lost after a nerve injury. The regenerated fibers showed a thinner myelin sheath and a reduction of the internodal distances [109,110,111]. It has been hypothesized that the hypomyelination of regenerated axons can be a result of an insufficient or dysregulated expression of neuronal growth factors—in particular, neuregulin-1 (NRG-1) [66].

In the presence of the axon, NRG-1 coupled with ErbB2/ErbB3 receptors on the SCs activate the pro-myelination signaling pathways, including the PI3K/PIP3/AKT pathway [6,112,113]. In vivo experiments in mice showed that an overexpression of ErbB2 induced a faster nerve regeneration after damage, as indicated by a faster motor recovery with the grasping test [114].

Of the three members of the neuregulin-1 family, NRG-1 type III plays an essential role in SC proliferation, differentiation and myelination during development and in the remyelination of regenerating nerves [24,46,48,115]. Interestingly, in vivo experiments in mice after severe crushed nerve lesions demonstrated that an overexpression of NRG-1 type III can restore the correct thickness of myelin. However, it can also induce hypermyelination [24,66]. In addition, this study demonstrated that, similarly to NRG-I type III, the overexpression of NRG1 type I also allows for the proper restoration of the myelin sheath of regenerating adult nerves. NRG-1 type I is produced and released by the regenerating axons. However, repair SCs can also produce NRG-1 type I via the autocrine mechanism. Of particular interest is the different behavior of SCs in response to the axonal NRG-1 type I or the self-produced NRG-1 type I. In the early phases of nerve regeneration characterized by the absence of axons and, therefore, the absence of NRG-1 type III and type I, the autocrine release of NRG-1 type I by the denervated SC favored their proliferation and migratory behavior and contributed to their morphological and functional cellular changes. Whereas, in the presence of regenerating axons, the paracrine release of axonal NRG-1 type I enriches the composition of the SC basal lamina and activates the intracellular mechanisms promoting remyelination [47]. In fact, in the experiments conducted in vivo by Stassart [66] in mutant mice with SCs lacking a functioning Nrg1 gene, the overexpression of either axonal NRG-1 type III or type I promoted nerve regeneration after nerve injury with a normal thickness of the myelin sheath. In conclusion, although sensitive to NRG-1 type I, the stimulus for SCs to initiate remyelination depends on the presence of axons.

## 6. Conclusions

SCs demonstrate remarkable plasticity when challenged by the environment of an injured peripheral nerve. SCs at the site of nerve lesions change morphologically and functionally to facilitate the repair process and then reestablish cell–axon contact in the final regenerative stage. The SC morphology is ultimately the result of forces generated by contact between the cells and their interactions with the ECM. The involvement of adhesion molecules generates traction of the membrane that affects the shape and movements of cells across the microenvironment. The repair SCs have the intrinsic ability to sense the biophysical changes in the microenvironment of the injured nerve and plastically modify their morphology and function. In this review, our goal was to concentrate on studies that analyze the physical conditions and properties that define the environment of an injured/regenerating nerve. We identified and described four phases of the process of peripheral nerve regeneration based on adaptive changes in SCs as a consequence of their mechanosensitivity that initiates adaptive genetic reprogramming. In reality, these phases are less distinct and more complex, with overlaps in time and space. SCs may simultaneously sense multiple physical constraints and respond with the prompt activation of signaling cascades to adequately react and proceed from a mature insulating supportive cell to an agile and multitasking cell of crucial importance for the regenerating axons. It has to be noted that the environmental conditions at the nerve lesion site may vary with the severity of the injury and contribute to the degree of regeneration of function recovery. The physical and chemical cues of the microenvironment at the site of the lesion that favor the degeneration and regeneration of axonal sprouts is compartmentalized within the preserved continuity of the nerve conduit, as seen in injuries like a neurapraxic insult or axonotmesis (types I-IV, according to Sunderland’s classification [116,117]). In the case of a complete nerve transection like neurotmesis (Sunderland Type V), the microenvironment favoring SC activation can be recreated within the autograft or allograft scaffolds bridging the nerve gap. Although peripheral nerve fibers can regenerate and function can be restored, a regenerated nerve greatly differs from an uninjured nerve. The difference is noticeable mostly in the increased ECM composition, while the cell composition and their proportions within the nerve are not affected [118]. Promising works in the fields of regenerative medicine and tissue engineering aim to enhance the quality of different types of biological and artificial nerve conduits to recreate the biochemical and biophysical conditions required for nerve regeneration.

## Figures and Tables

**Figure 1 cells-10-00425-f001:**
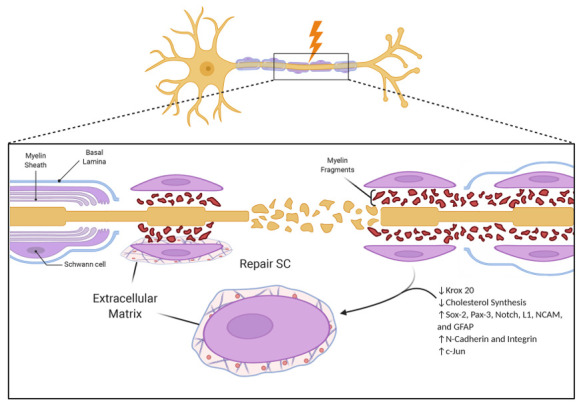
The loss of contact with the axon activates morpho-functional changes leading to the repair Schwann cells (SCs). The figure was adapted from “Peripheral Nervous System (PNS) Myelin Structure” by BioRender.com (accessed on 2020). GFAP, Glial fibrillary acidic protein.

**Figure 2 cells-10-00425-f002:**
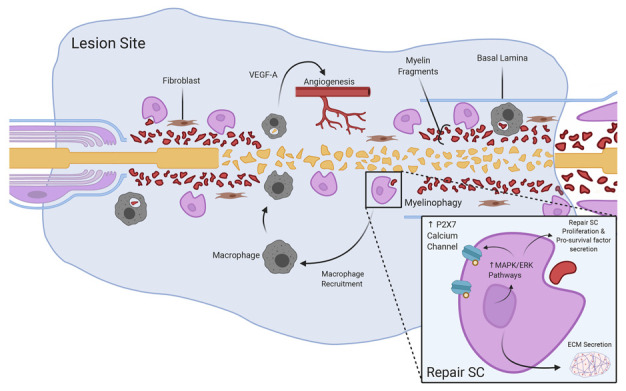
Cell recruitment to the lesion site and the secretion of pro-survival factors, angiogenic factors and fibrils. The figure was adapted from “Peripheral Nervous system (PNS) Myelin Structure” by BioRender.com (accessed on 2020). SC, Schwann cell; ECM, extracellular matrix; VEGF-A, vascular endothelial growth factor-A.

**Figure 3 cells-10-00425-f003:**
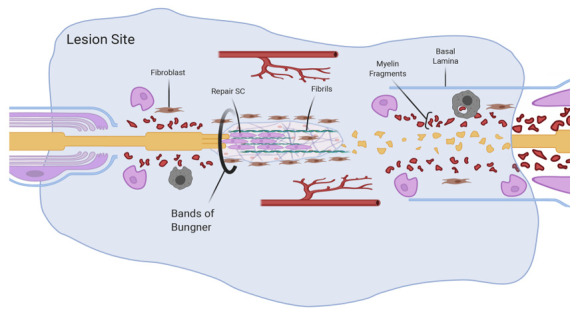
Schwann cells migrate onto a new matrix and onto cells and tissue, which confers directionality. The figure was adapted from “Peripheral Nervous System (PNS) Myelin Structure” by BioRender.com (accessed on 2020).

**Figure 4 cells-10-00425-f004:**
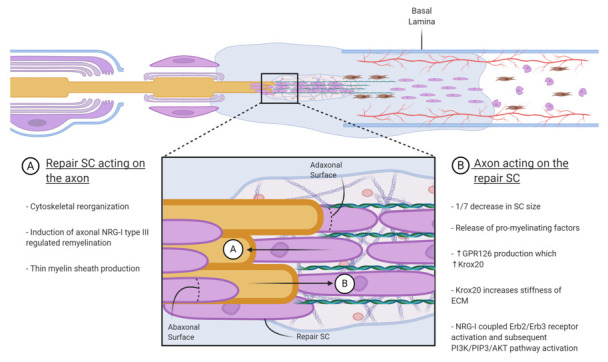
The contact with regrowing axons initiates remyelination. The figure was adapted from “Peripheral Nervous System (PNS) Myelin Structure” by BioRender.com (accessed on 2020). NRG-I, Neuregulin-I: ECM, extracellular matrix; GPR126, G protein-coupled receptor 126.

## Data Availability

Not applicable.

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
