# Peer review of "Dynamic Environmental Physical Cues Activate Mechanosensitive Responses in the Repair Schwann Cell Phenotype"

_cells, 2021, doi:10.3390/cells10020425_

Round 1

Reviewer 1 Report

Review of

Dynamic Environmental Physical Cues Activate Mechanosensitive Responses in the Repair Schwann Cell Phenotype

This review focusses on evidence on physical cues and how these affects the nerve regeneration. It nicely summarises these evidence and is separated into 4 sections. I found the review very interesting and well done.

Here are some critique:  

During the description -  Phase 2 and Phase 3, there is a change from passive form to active. 

Lines 57 & 60: “In this new approach… With this new approach…”. Is this really needed here? The previous sentence contains such an expression so please reconsider.

Fig.1: I would suggest at least an annotation of the myelin fragments and I assume of the blue line which is the basal lamina. Maybe change the colour of the myelin fragments in order to show that they are a product of Schwann cells…it might help in the next figure where there is myelinophagy.

Line 94 Fig.1 legend: “…SC phenoTable 2020.” What is this?

Lines 97 & 98: “…new molecular mechanisms will lead to morphological…” Should be… new molecular mechanisms, leading to morphological…

Line 104 and 105: Generally speaking yes, but if you consider the downregulation of Krox20, cholesterol and myelin genes as an adaptive reprogramming, then obviously the non-myelinating SCs differ.

Line 105 & 106: “The reprogramming is driven by a combination of dedifferentiating and differentiating programs.”  So if all the changes that are described afterwards are involved in the dedifferentiation program, what processes of differentiation are involved in the adaptive reprogramming?

Lines 138-147: Firstly, I believe these two paragraphs should be one.

There is not a mention of why the role of Neuregulin 1 is controversial.

While not in the scope of this review, I believe further evidence should be provided. I believe the following papers also provide supporting evidence for the role of Neuregulin 1 during PNS injury. Fledrich et al., 2019, Strassart et al., 2013, Syed et al., 2010, Huijbregts et al., 2003). The axonal role of neuregulin 1: Fricker et al., 2013.

With so many types of neuregulin and so many different cell types expressing them, it is of no surprise that there is controversy. Strassart et al., 2013 and the Neuregulin contribution is discussed in more detail in Phase Four. Maybe the authors need to refer to that section.

There is also controversy about the ErbB2/3 and the intracellular signalling cascade and also the role of MAPK/ERK. Sheean et al., 2014 proved that the MAPK/ERK signalling is also involved in myelination, not only suppression of myelination. Sustained activation of ERK impairs nerve repair (Cervellini 2018). And traditionally the ErbB2/3 signalling activates the PI3K -Akt pathway for the myelination. (Reviewed in Birchmeier and Bennett 2016).

Line 162 Fig. 2: I would suggest the macrophages to be a different colour. It might be a different purple than the Schwann cell but I would prefer more distinction. As with the previous picture, the basal lamina and the myelin fragments should be labelled. It might be useful to colour the myelin a different purple from the Schwann cell. In this case the myelin fragment can be visualised inside the SC and macrophage cytoplasm to show the myelinophagy.

Line 227 Fig. 3: Similar comments to the figures before.

Line 258: “…reported that GGF-treated SCs…”. What does this abbreviation stand for?

Line 282: “…involves secretion of NGF, BDNF and CNTF into the ECM…” Please provide first the full name or create a list of abbreviations.

Line 317: “…that the topography affected orientation of the…” Should be… that the topography affected the orientation of the…

Line 327: “…The repair SCs had different size…” The repair SCs have different size…

Lines 347 & 348: Knox20 should be changed to Krox20

Line 360: What you mean with irregularities in the distribution of internodes is a generic reduction in the internodal distances which is due to the SC proliferation. Since more Schwann cells have to fit in the regenerated area, the changes in the internodal distances are not surprising.

It should also be noted here that despite the mentioned changes before and after injury, the cell composition and their proportion within the nerve is not affected. (Stierli et al., 2018) 

Author Response

We thank the reviewer for the thoughtful feedback to our manuscript. In accordance with the suggestions we edited the manuscript as follow:

R: During the description - Phase 2 and Phase 3, there is a change from passive form to active. 

A: We made some changes in the use of verbs in the text and we believe we address this issue.

R: Lines 57 & 60: “In this new approach… With this new approach…”. Is this really needed here? The previous sentence contains such an expression so please reconsider.

A: We agree with the redundancy and modified the text (line 60)

R: Fig.1: I would suggest at least an annotation of the myelin fragments and I assume of the blue line which is the basal lamina. Maybe change the colour of the myelin fragments in order to show that they are a product of Schwann cells…it might help in the next figure where there is myelinophagy.

R: Line 162 Fig. 2: I would suggest the macrophages to be a different colour. It might be a different purple than the Schwann cell but I would prefer more distinction. As with the previous picture, the basal lamina and the myelin fragments should be labelled. It might be useful to colour the myelin a different purple from the Schwann cell. In this case the myelin fragment can be visualised inside the SC and macrophage cytoplasm to show the myelinophagy.

R: Line 227 Fig. 3: Similar comments to the figures before.

A: We believe the suggested changes improved the quality of all Figures. We made the changes suggested.

R: Line 94 Fig.1 legend: “…SC phenoTable 2020.” What is this?

A: We apologize for the typos. The line was removed and the legend reworded (Line 95)

R: Lines 97 & 98: “…new molecular mechanisms will lead to morphological…” Should be… new molecular mechanisms, leading to morphological…

A: We thank for the suggestion. The sentence was reworded as per lines 97-100.

R: Line 104 and 105: Generally speaking yes, but if you consider the downregulation of Krox20, cholesterol and myelin genes as an adaptive reprogramming, then obviously the non-myelinating SCs differ.

A: We agree and modify the text as per lines 107-118

R: Line 105 & 106: “The reprogramming is driven by a combination of dedifferentiating and differentiating programs.”  So if all the changes that are described afterwards are involved in the dedifferentiation program, what processes of differentiation are involved in the adaptive reprogramming?

A: The repair cell is the result of a reprogramming leading to a dedifferentiation and redifferentiation into a new cell different from the neonate SCs of developmental stages. The differentiation includes the de novo expression of molecular markers like Olig1, c-Jun, Shh and the downstream signaling pathways associated with these molecules. 

R: Lines 138-147: Firstly, I believe these two paragraphs should be one.

A: we agree with the reviewer and combined the two paragraphs (lines 141-155).

R: While not in the scope of this review, I believe further evidence should be provided. I believe the following papers also provide supporting evidence for the role of Neuregulin 1 during PNS injury. Fledrich et al., 2019, Strassart et al., 2013, Syed et al., 2010, Huijbregts et al., 2003). The axonal role of neuregulin 1: Fricker et al., 2013.

With so many types of neuregulin and so many different cell types expressing them, it is of no surprise that there is controversy. Strassart et al., 2013 and the Neuregulin contribution is discussed in more detail in Phase Four. Maybe the authors need to refer to that section.

There is also controversy about the ErbB2/3 and the intracellular signalling cascade and also the role of MAPK/ERK. Sheean et al., 2014 proved that the MAPK/ERK signalling is also involved in myelination, not only suppression of myelination. Sustained activation of ERK impairs nerve repair (Cervellini 2018). And traditionally the ErbB2/3 signalling activates the PI3K -Akt pathway for the myelination. (Reviewed in Birchmeier and Bennett 2016).

A: We agree with the reviewer and we reworded the sentence avoiding the term “controversy” mostly because as per many other factors, the many types of neuregulin play multiple roles often leading to opposite results depending on the circumstances. As suggested by the reviewer, we refer to phase 4 where we discuss neuregulin roles with more details. We also added the references suggested and a recent review article on this topic (Stassart, RM, Developmental neurobiology 2020) in phase 4 which discusses the remyelination process.

R: Line 258: “…reported that GGF-treated SCs…”. What does this abbreviation stand for?

A: GGF stand for “Glial growth factor. We spelled it out at now line 270.

R: Line 282: “…involves secretion of NGF, BDNF and CNTF into the ECM…” Please provide first the full name or create a list of abbreviations.

A: We provided full names of the factors as suggested.

R: Line 317: “…that the topography affected orientation of the…” Should be… that the topography affected the orientation of the…

A: We edited the text as suggested (line 330).

R: Line 327: “…The repair SCs had different size…” The repair SCs have different size…

A: We apologize for the grammatical error. The text was edited as suggested (line 340).

R: Lines 347 & 348: Knox20 should be changed to Krox20

A: We apologize for the typo. Changes were made through the text.

R: Line 360: What you mean with irregularities in the distribution of internodes is a generic reduction in the internodal distances which is due to the SC proliferation. Since more Schwann cells have to fit in the regenerated area, the changes in the internodal distances are not surprising.

A: We agree the term irregularities was misleading and changed the sentence in …” reduction of internodal distance” (line 371).

R: It should also be noted here that despite the mentioned changes before and after injury, the cell composition and their proportion within the nerve is not affected. (Stierli et al., 2018) 

A: We agreed and added a statement in the discussion (lines 428-431).

Reviewer 2 Report

I found the review interesting and well written. I only have a few comments:

  • As a surgeon I would find it interesting if there is literature on potential differences in the microenvironments for different types of nerve injury and repair. It now seems like all four phases are similar for different types of nerve injury, while the literature references are based on different types of injury and repair (f.e. crush injury versus transection and repair). For example the formation of the fibrin matrix is based on research performed in nerve tubes (reference 88). Does the same formation occur in direction coaptation repair? I would be interesting to discuss the role of Schwann cells in autografts vs decellularized allografts and the limitation of the latter in bridging of large defects. Also interesting to discuss potential differences in Schwann cells for interaction with motor versus sensory axons. This would of course increase the length of the text and number of references, but I now find that article discuss different intracellular pathways quite extensive. This could be limited. Also, authors can sometimes refer to other reviews, for example in the part discussing nerve growth factors (ref 65-73), which limits the number of references.

  • It also would be nice if the authors provided a timeline for the different phases and illustrate it with different pictures of Schwann cells, somewhat similar to figure 1 in review Jesser and Mirsky (reference 20).

Otherwise enjoyed reading this manuscript. Best regards,

Author Response

We thank the reviewer for his thoughtful comments to our manuscript. Please find below our responses:

R: I found the review interesting and well written.

A: thank you for the support and encouragement.

R: I only have a few comments:

  • As a surgeon I would find it interesting if there is literature on potential differences in the microenvironments for different types of nerve injury and repair. It now seems like all four phases are similar for different types of nerve injury, while the literature references are based on different types of injury and repair (f.e. crush injury versus transection and repair). For example, the formation of the fibrin matrix is based on research performed in nerve tubes (reference 88). Does the same formation occur in direction coaptation repair? I would be interesting to discuss the role of Schwann cells in autografts vs decellularized allografts and the limitation of the latter in bridging of large defects. Also, interesting to discuss potential differences in Schwann cells for interaction with motor versus sensory axons. This would of course increase the length of the text and number of references, but I now find that article discuss different intracellular pathways quite extensive. This could be limited. Also, authors can sometimes refer to other reviews, for example in the part discussing nerve growth factors (ref 65-73), which limits the number of references.

A: I totally understand the interest of the reviewer as I worked myself with surgeons and investigated nerve regeneration using different injury models from axonotmesis, to crush or neurotmesis. I believe the topic suggested by the review could be discussed in full in a different manuscript. However, if covered in our manuscript, it would change the aim we chose for this work. This review is aimed at defining the influence of the physical and chemical cues of the microenvironment at the lesion site in the transformation of the SC into the repair phenotype. I agree with the reviewer that the microenvironment can differ with different type of lesion and therefore we added a paragraph in discussion to acknowledge the and clarify this point. As the reviewer suggested, this review took a different direction analyzing more in depth the molecular pathways activated as the result of Schwann cells mechanosensitivity. We focused on how these multipotent cells can change their phenotype and functions in response to physical cues of the environment, some of which (like the presence or not of the axons or the recruitment of other cell types) are probably similar upon all the different types of injuries.

  • It also would be nice if the authors provided a timeline for the different phases and illustrate it with different pictures of Schwann cells, somewhat similar to figure 1 in review Jesser and Mirsky (reference 20).

A: This is a good suggestion and we tried to address the morphological changes of SC in Fig 1,3,4 where we depicted how the myelinating cells after losing contact with the axon, changes morphology (fig.1) and as a consequence of the readapting programs acquires secreting functions (fig.3). Its morphology changes again in presence of regenerating axons (Fig.4).

Otherwise enjoyed reading this manuscript. Best regards,

A: Thank you.